# Relationship between NAFLD and Periodontal Disease from the View of Clinical and Basic Research, and Immunological Response

**DOI:** 10.3390/ijms22073728

**Published:** 2021-04-02

**Authors:** Masahiro Hatasa, Sumiko Yoshida, Hirokazu Takahashi, Kenichi Tanaka, Yoshihito Kubotsu, Yujin Ohsugi, Takaharu Katagiri, Takanori Iwata, Sayaka Katagiri

**Affiliations:** 1Department of Periodontology, Graduate School of Medical and Dental Sciences, Tokyo Medical and Dental University (TMDU), Tokyo 113-8549, Japan; hatasa.peri@tmd.ac.jp (M.H.); sumiperi@tmd.ac.jp (S.Y.); ohsugi.peri@tmd.ac.jp (Y.O.); iwata.peri@tmd.ac.jp (T.I.); 2Division of Metabolism and Endocrinology, Faculty of Medicine, Saga University, Saga 849-8501, Japan; sj8833@cc.saga-u.ac.jp (K.T.); y.05211027@gmail.com (Y.K.); 3Liver Center, Saga University Hospital, Faculty of Medicine, Saga University, Saga 849-8501, Japan; 4Department of Biochemistry, Toho University School of Medicine, Tokyo 143-8540, Japan; takaharu.katagiri@med.toho-u.ac.jp; 5Division of Rheumatology, Department of Internal Medicine, Ohashi Medical Center, Tokyo 153-8515, Japan

**Keywords:** periodontal disease, NAFLD, microbiome, inflammation, immunological response

## Abstract

Periodontal disease is an inflammatory disease caused by pathogenic oral microorganisms that leads to the destruction of alveolar bone and connective tissues around the teeth. Although many studies have shown that periodontal disease is a risk factor for systemic diseases, such as type 2 diabetes and cardiovascular diseases, the relationship between nonalcoholic fatty liver disease (NAFLD) and periodontal disease has not yet been clarified. Thus, the purpose of this review was to reveal the relationship between NAFLD and periodontal disease based on epidemiological studies, basic research, and immunology. Many cross-sectional and prospective epidemiological studies have indicated that periodontal disease is a risk factor for NAFLD. An in vivo animal model revealed that infection with periodontopathic bacteria accelerates the progression of NAFLD accompanied by enhanced steatosis. Moreover, the detection of periodontopathic bacteria in the liver may demonstrate that the bacteria have a direct impact on NAFLD. Furthermore, *Porphyromonas gingivalis* lipopolysaccharide induces inflammation and accumulation of intracellular lipids in hepatocytes. Th17 may be a key molecule for explaining the relationship between periodontal disease and NAFLD. In this review, we attempted to establish that oral health is essential for systemic health, especially in patients with NAFLD.

## 1. Introduction

Periodontal disease, especially chronic periodontitis, is an infectious disease induced by oral bacteria that can lead to the destruction of soft tissues surrounding the teeth, bones, and ligaments [1]. Bacteria in plaques are closely involved with the onset of periodontal disease, and the mucosal epithelium is inflamed by exotoxins produced by the bacteria [2]. Periodontal disease results in not only tooth loss but also the aggravation of numerous types of systemic diseases, including type 2 diabetes [3,4], cardiovascular diseases [5,6], preterm low birth weight [7], and nonalcoholic fatty liver disease (NAFLD) [8,9,10]. Thus, monitoring and management of periodontitis is important because it is present in almost half the adult population [11]. There are three hypotheses for how periodontitis affects systemic diseases. Firstly, local periodontal infection leads to an increase in systemic inflammatory mediators [12,13]. In fact, some studies have shown that increased inflammatory mediator levels were decreased after successful periodontal treatment [14]. Secondly, in patients with periodontitis, bacteremia may originate from periodontal pockets, as patients with generalized chronic periodontitis showed ulcers in inflamed periodontal pockets, and the total area of the ulcers was estimated to be as large as the size of the palm [15]. Thirdly, alterations in the oral microbiome due to periodontal disease may affect the gut microbiome. Salivary levels of *Aggregatibacter actinomycetemcomitans*, *Porphyromonas gingivalis*, and *Prevotella intermedia* were determined by bacterial culture and related to clinical periodontal status in subjects with varying degrees of periodontitis [16]. The gastrointestinal tract begins with the mouth and proceeds to the intestines; ingested bacteria travel through the tract; thus, affect gut microbiota composition [17]. Dysbiosis of the gut microbiota can lead to several diseases, including diabetes, rheumatoid arthritis, and inflammatory bowel disease [18].

The purpose of this review was to assess the relationship between periodontal disease and NAFLD through evaluating epidemiological and interventional studies in humans, studies of periodontitis model animals, and in vitro studies. We aimed to provide information that could contribute to the future development of basic and clinical research regarding the relationship between NAFLD and periodontal disease.

## 2. NAFLD

### 2.1. NAFLD and Nonalcoholic Steatohepatitis (NASH)

NAFLD is a clinical entity characterized by the presence of hepatic steatosis affecting at least 5% of hepatocytes in individuals who consume little or no alcohol and who do not have a secondary cause of hepatic steatosis [19]. NAFLD includes heterogeneous spectra ranging from simple steatosis to NASH and liver cirrhosis [20]. Liver fibrosis is an independent risk factor affecting the prognosis of patients with NASH [21], and a systematic review and meta-analysis concluded that liver fibrosis was the most important liver histological finding for all-cause and liver disease-related mortality in NAFLD [22]. A meta-analysis of the global epidemiology of NAFLD reported that its prevalence has increased in the last 10 years, affecting 26.8% of the global population [23]. Therefore, it is important to identify patients with NAFLD and advanced NASH in daily practice and to prompt lifestyle interventions. To prevent the progression of NASH and liver fibrosis, more than 300 trials for new drugs were ongoing in 2018 [24], but effective pharmacological therapy is currently lacking.

The main causes of NAFLD are the exacerbation of insulin resistance concomitant with obesity, type 2 diabetes, dyslipidemia, and hypertension. In particular, type 2 diabetes is strongly associated with the development and progression of NAFLD [25,26]. The two-hit theory has been advocated as the etiological pathogenesis of NAFLD/NASH; obesity, overnutrition, and insulin resistance were considered the first hits and cause hepatic steatosis, and the second hits, including dyshomeostasis of oxidative stress, gut-derived endotoxins, and free fatty acids, develop liver steatosis to NASH [27]. However, the “multiple parallel hits hypothesis” was advocated later [28]. Various factors involved in the development of inflammation and fibrosis may act in parallel with the liver and finally develop into NASH. The interaction between the liver and other organs, such as the adipose tissue and intestines, leads to the progression of NAFLD/NASH; specifically, increased oxidative stress due to increased lipid influx to hepatocytes, promotion of insulin resistance, abnormal secretion of adipocytokines from adipose tissue, and endotoxin influx from the intestinal tract may be associated with the simultaneous development and progression of NAFLD/NASH [28]. Furthermore, diseases of the oral environment, such as periodontal disease, are considered factors affecting NAFLD.

The direct effects of periodontal disease on the clinical pathogenesis of NAFLD and NASH are not fully understood. However, multiple cross-sectional studies have shown the association between infection by periodontitis-associated bacteria and the hepatic phenotypes of human NAFLD. That said, the significance of periodontitis in the clinical outcomes of NAFLD and NASH, including mortality, liver cancer, liver failure, and cardiovascular disease events, remains unclear. A longitudinal randomized controlled study should be conducted for evaluating the effect of periodontitis treatment on NAFLD and NASH in humans.

### 2.2. From NAFLD to Metabolic Associated Fatty Liver Disease (MAFLD)

NAFLD was initially named by Schaffner in 1986 [29], but according to the international consensus panel of the American Gastroenterological Association (AGA), a consensus-driven proposed nomenclature for “metabolic associated fatty liver disease (MAFLD)” was issued in 2020.

The heterogeneity in the clinical presentation and course of fatty liver disease is influenced by a multitude of factors, including age, sex, ethnicity, alcohol intake, dietary habits, hormonal status, genetic predisposition, epigenetic factors, microbiota, and metabolic status. It is likely that there is a differential impact on the contribution of the various factors in any individual over time and among individuals that then shapes the disease phenotype and course [30].

Eslam et al. suggested that MAFLD is a more appropriate overarching term and provides new terminology that more accurately reflects pathogenesis and can help in patient stratification for management. MAFLD can be diagnosed if either obesity or type 2 diabetes is diagnosed concomitantly with fatty liver disease. MAFLD can also be diagnosed in lean or non-obese patients with fatty liver if they have at least two metabolic abnormalities, including dyslipidemia, hypertension, insulin resistance, and prediabetes. Moreover, other chronic liver diseases, such as viral hepatitis, autoimmune diseases, or alcoholic liver injury, are not an exclusion criteria for MAFLD diagnosis [30]. To the best of our knowledge, the association between MAFLD and periodontal disease has not been reported previously. Considering the broader spectrum of MAFLD, which focuses on metabolic abnormalities and can occur in patients with any other chronic liver disease, the association between periodontal disease and MAFLD may be significant and more complicated than that with NAFLD. Type 2 diabetes and metabolic syndrome are the sole risk factors for periodontal disease [31], and the pathogenesis of viral hepatitis and alcoholic liver injury have also been linked to periodontal disease [32]. Further studies are necessary to identify the clinical significance of periodontal disease in patients with MAFLD. While this proposed update to the MAFLD nomenclature is expected to accelerate the development of new biomarkers and drugs, Younossi et al. recommended that it is important not to rush into a name change and experts should be cautious to prevent confusion because the definition of the term remains ambiguous [33]. On the other hand, regardless of the definition of the “fatty liver,” either NAFLD or MAFLD, it is important to understand the underlying mechanism that aggravates the fatty liver to inflammation and fibrosis in association with concomitant diseases and pathogenesis, considering fatty liver is the manifestation of systemic disease association.

## 3. Human Epidemiologic Studies

### 3.1. Cross-Sectional Studies Regarding the Relationship between NAFLD and Periodontal Disease

Since 2010, epidemiological studies regarding the relationship between NAFLD and periodontal disease have attracted the interest of many research groups worldwide. Many investigations have been performed in East Asia. The first study regarding the relationship between liver disease and periodontal disease was conducted in Japan by Furuta et al., reporting that having periodontitis was significantly associated with elevated serum alanine aminotransferase (ALT) levels in 2225 non-smoking male university students (OR = 2.3, 95% CI = 1.0–5.2) [34]. Several research groups in Japan have revealed further associations. A similar study was conducted by assessing the health examinations of 5683 adults. A positive correlation between deep periodontal pockets and the combination of increased serum ALT and symptoms of metabolic syndrome was observed in male subjects with low alcohol consumption, but not in female [35]. Furthermore, an association between the elevation of γ-glutamyltranspeptidase and deep periodontal pockets (OR = 1.48, 95% CI = 1.16–1.9) was reported in 1510 adults [36]. After adjusting for confounding factors, the odds ratio of having PD ≥ 4 mm for NAFLD was still significant in 1226 adults (OR = 1.881, 95% CI = 1.184–2.987). In all participants and in female participants, the prevalence of NAFLD increased with the increase in periodontal disease severity [37]. In Korea, two studies demonstrated associations between the fatty liver index (FLI) and periodontal disease in all subjects and only in women by analyzing data from a national health survey of more than 4000 people [38,39]. A stronger association was observed between FLI and periodontitis prevalence in the diabetes subgroup. In addition, patients with diabetes showed more severe periodontitis [38]. The severity of periodontal disease was positively associated with FLI in female participants [39]. A large-scale investigation was conducted in China. More than 24,000 individuals were screened for NAFLD by ultrasonography (USON), and there was a significant correlation between the number of missing teeth and NAFLD in men [40].

For other regions, two studies reported results from the National Health and Nutrition Examination Survey in the USA. USON, fibrosis score (FS), FLI, the United States fatty liver index (US-FLI), and the correlation between periodontitis and tooth loss were examined in 5421 adults. In adjusted models, adults with moderate-severe periodontitis were more likely to have NAFLD (USON: OR = 1.54, 95% CI = 1.06–2.24; FS: OR = 3.10, 95% CI = 2.31–4.17; FLI: OR = 1.61, 95% CI = 1.13–2.28; US-FLI: OR = 2.21, 95% CI = 1.64–2.98). People with <20 teeth were also more likely to have NAFLD compared with those with ≥20 teeth (USON: OR = 1.50, 95% CI = 1.11–2.02; FS: OR = 4.36, 95% CI = 3.47–5.49; FLI: OR = 1.99, 95% CI = 1.52–2.59; US-FLI: OR = 2.32, 95% CI = 1.79–3.01) [41]. In another study, an association between periodontitis and hepatic steatosis diagnosed by hepatic ultrasound data from 8172 people was examined. The odds ratio for steatosis was statistically significant for bleeding on probing (BOP), probing depth (PD) ≥ 4 mm, mean PD, clinical attachment level (CAL) ≥ 3 mm, and mean CAL (%). After adjusting for sociodemographic factors, only BOP and mean PD showed a significant association with steatosis (BOP: OR = 1.07, 95% CI = 1.00–1.14, mean PD: OR = 1.08, 95% CI = 1.00–1.17). Associations between NAFLD and serum antibacterial antibody titers against oral bacteria in 3236 adults who had USON-proven hepatic steatosis and antibody titers against *Selenomonas noxia* and *Streptococcus oralis* were found to be significant. There was also a weak association between hepatic steatosis and cluster score from antibody titers against bacterial complexes (*Tannerella forsythia*, *Treponema denticola*, *A. actinomycetemcomitans mix*), which is related to severe periodontitis, and four bacteria (*Eikenella corrodens*, *S. noxia*, *Veillonella parvula*, and *Campylobacter rectus*) which contribute to the early stage of plaque accumulation [42]. Furthermore, a study focused on a high-risk group for NAFLD was conducted, targeting 11,914 Hispanics and Latinos, in the USA. Periodontitis was associated with serum ALT and aspartate aminotransferase (AST) levels (≥30% of sites with PD ≥ 4 mm: OR = 1.39, 95% CI = 1.02–1.90); however, the significance was lost after adjustment for age and sex [43]. Additionally, the same research group examined 2481 Germans to analyze the relationships between serum CRP levels, periodontitis, and NAFLD. The relationships among the weighted genetic CRP score, which combines serum CRP levels and single-nucleotide polymorphisms previously identified through genome-wide association studies as robustly associated with serum CRP [44,45], periodontitis, and NAFLD, were investigated. A significant correlation between periodontitis and NAFLD was found among those with <1 mg/L serum CRP levels and/or with lower than the median weighted genetic CRP score. Interestingly, serum CRP levels modified the interaction between periodontitis and NAFLD. Subjects with extensive periodontitis (≥30% sites with PD ≥ 4 mm) had higher odds ratios for NAFLD than patients with moderate periodontitis (<30% sites with PD ≥ 4 mm) among participants with a serum CRP level <1 mg/L [46].

Many epidemiological studies have supported the relationship between NAFLD and periodontal disease. In addition, compared to moderate periodontal disease, severe periodontal disease is more strongly associated with NAFLD. However, both diseases are strongly related to metabolic disorders, and this relationship should be carefully considered. The contents of this section are summarized in Table 1.

### 3.2. Prospective Cohort Studies Regarding the Relationship between NAFLD and Periodontal Disease

Prospective cohort studies regarding the relationship between periodontitis and NAFLD were conducted by three research groups. Kuroe et al. monitored the incidence of NAFL in 4812 study participants over five years and analyzed its association with periodontitis. In total, 341 individuals were diagnosed with NAFL during the observational period. CAL, which indicates the extent of distracted periodontal tissue, and liver fibrosis were significantly associated in obese patients with NAFL (OR = 2.87, 95% CI = 1.23–6.69) [47]. In a German population, 2623 individuals were followed up for a median of 7.7 years to assess the relationship between periodontitis and NAFLD onset. NAFLD incidence was elevated in participants with >30% of ≥3 mm CAL (multivariable-adjusted incidence rate ratio: 1.60, 95% CI = 1.05–2.43). However, NAFLD and periodontitis were not associated in participants with <30% affected sites. Surprisingly, the incidence rate ratio of the change in CAL in five years significantly increased upon combination with NAFLD in participants with >30% of ≥3 mm CAL, whereas the significance was not observed after adjustment for confounding factors. NAFLD may contribute to the aggravation of periodontitis [48]. The longest follow-up study was conducted over 13 years in Finland. A total of 1801 patients with NAFLD were followed up to assess the relationship between periodontitis and the incidence of severe liver disease. The incidence of severe liver disease was defined as follows: first hospitalization due to liver disease or liver-related death, a diagnosis of (primary) liver cancer, whichever came first. The hazard ratio for the incidence of severe liver disease increased with the severity of periodontitis. Interestingly, the hazard ratio of the incidence of severe liver disease was elevated to 6.94 for patients with advanced periodontitis and NAFLD (95% CI = 1.43–33.6). These results suggest that periodontitis contributes to the aggravation of NAFLD [49]. The contents of this section are summarized in Table 2.

### 3.3. Meta-Analysis Regarding the Relationship between NAFLD and Periodontal Disease

Recently, two systematic reviews and meta-analysis regarding the relationship between periodontal disease and NAFLD have been published. Chen et al. reported that periodontitis was associated with NAFLD and cirrhosis by reviewing 12 studies. Among patients with periodontal disease, the odds ratio for NAFLD was 1.19 (95% CI = 1.06–1.33) and that for cirrhosis was 2.28 (95% CI = 1.50–3.48). In addition, tooth loss was associated with NAFLD (OR = 1.33, 95% CI = 1.12–1.56). [50]. Wijarnpreecha et al. analyzed over five studies (27,703 subjects) and found that periodontitis was associated with NAFLD (OR = 1.48, 95% CI = 1.15–1.89). However, no significant correlation between periodontitis and NAFLD was observed after the adjusted results were applied in the meta-analysis from the primary studies. The confounding factors, including various metabolic parameters, were adjusted [51]. The contents of this section are summarized in Table 3.

### 3.4. Reports Regarding Periodontal Disease in Patients with NAFLD

The effects of periodontitis on NAFLD severity were assessed in five cross-sectional NAFLD patient-based studies. The relationships among periodontitis, diabetes, and severity of NAFLD were investigated in 69 patients with NAFLD. Periodontitis is significantly associated with the severity of NAFLD (NASH vs. NAFL, *p* = 0.0085) and the presence of diabetes [42]. Microbiological analyses have been conducted as a parameter of periodontitis in several studies. Yoneda et al. examined the relationship between NAFLD and the presence of *P. gingivalis,* one of the major periodontopathic bacteria, and found that the detection of *P. gingivalis* was significantly higher in patients with NAFLD than in non-NAFLD control subjects. Half of the detected *P. gingivalis* from patients with NAFLD was categorized as *fim A* type 2 [8]. *Fim A* type 2 *P. gingivalis* feature strong adhesion and invasion to host cells and are associated with severe periodontitis [52,53]. Serum anti-*P. gingivalis* antibody titers and liver biopsy findings were investigated in 200 patients with NAFLD. A significant monotonic trend was observed between the fibrosis severity and antibody titers against *P. gingivalis fim A* type 1 and 4. In addition, antibody titers against *P. gingivalis fim A* type 4 were associated with advanced fibrosis in multivariate analysis (OR = 2.081, 95% CI = 1.098–3.943) [54]. *P. gingivalis* was detected by immunohistochemistry in hepatocytes from liver biopsy specimens of 40 patients with NAFLD. *P. gingivalis*-positive patients showed progression of hepatic fibrosis compared to patients without *P. gingivalis* [55]. Furthermore, we conducted clinical research based on the concept that periodontopathic bacteria may aggravate NAFLD by affecting lipid and glucose metabolism. Komazaki et al. focused on metabolic parameters in NAFLD patients to assess the effect of periodontopathic bacteria. Serum antibody titers against three major periodontopathic bacteria (*A. actinomycetemcomitans*, *Fusobacterium nucleatum*, and *P. gingivalis*) were examined, and their correlations with metabolic parameters were evaluated. Anti-*A. actinomycetemcomitans* antibody titers and anti-*F. nucleatum* antibody titers were slightly associated with fat area, evaluated by abdominal computed tomography. Anti-*A. actinomycetemcomitans* antibody titers also showed a significant correlation with fasting plasma insulin, the homeostasis model of assessment of insulin resistance, and AST. Moreover, anti-*A. actinomycetemcomitans* antibody titers were negatively correlated with the liver/spleen ratio evaluated by abdominal computed tomography [9].

Up to the present, only one interventional study has investigated the effect of periodontal treatment on NAFLD. Ten NAFLD patients received non-surgical periodontal treatment for three months. Serum AST and ALT levels were significantly decreased after periodontal treatment [8]. The contents of this section are summarized in Table 4.

## 4. In Vivo Basic Research Regarding the Relationship between NAFLD and Periodontal Disease

Several studies have investigated the effects of periodontal disease on NAFLD in vivo. In this section, we reviewed these reports according to each experimental model. The contents of this section are summarized in Table 5.

### 4.1. P. gingivalis Lipopolysaccharide (LPS) Injection in Gingiva Model

Rats fed a high-fat diet for 12 weeks and injected with *P. gingivalis* LPS in the gingiva once a day for 10 days showed large fat droplets, ballooning degeneration, and infiltration of inflammatory cells. These results demonstrated that high-fat diet feeding and *P. gingivalis*-LPS injection accelerated the progression from simple steatosis to NASH in the liver. Moreover, high radioactivity was observed in the liver of rats injected with double-radiolabeled *P. gingivalis* LPS at 24 h [56]. Rabbits fed a high-fat diet for 40 days and injected *P. gingivalis* LPS in the gingiva twice a week had a high score of acinar inflammation in the liver and increased blood triglyceride and phospholipid levels compared to rabbits without injected *P. gingivalis* LPS [57].

### 4.2. Pulp Chamber Model

When *P. gingivalis* is injected into the pulp cavity, it is cultured in the pulp cavity and the body receives a continuous supply of *P. gingivalis*. *P. gingivalis* is transferred from the pulp cavity to the body through the root apex of the teeth [55], and the model can evaluate how continuous *P. gingivalis* infection affects the systemic organs.

Mice were fed high-fat diet for 12 weeks and then infected with *P. gingivalis* from the pulp chamber. These mice showed increased serum levels of LPS and more foci of Mac2-positive macrophages. Furthermore, *P. gingivalis* was detected in Kupffer cells and hepatocytes at 6 weeks after *P. gingivalis* infection [55]. Fibrosis and steatosis were more severe in the livers of mice fed high-fat diet for 12 weeks and infected with *P. gingivalis* compared to mice without *P. gingivalis* infection. Metabolome analysis of the liver was performed using CE-TOFMS and LC-TOFMS. Fatty acid metabolism was significantly disrupted, and expression levels of SCD1 and ELOVL6 were significantly reduced in *P. gingivalis*-infected mice [54]. In addition, the same model mice showed an increased number of hepatic crown-like structures, which was macrophage aggregation and related to liver fibrosis, and the fibrosis area was also increased by upregulating the immunoexpression of phosphorylated Smad2 (a key signaling molecule of TGF-β1) and Galectin-3 at 9 weeks after *P. gingivalis* infection. *P. gingivalis* was detected in the liver by immunohistochemistry [58].

### 4.3. Intravenous Injection of the P. gingivalis Model

A single intravenous injection of *P. gingivalis* increased the bodyweight and accelerated the progression of NAFLD in mice fed high-fat diet for 4 weeks [8]. Endotoxemia induced by intravenous injection of sonicated *P. gingivalis* twice a week caused impaired glucose tolerance, insulin resistance, and liver steatosis in mice fed high-fat diet for 12 weeks. Liver microarray analysis demonstrated that fatty acid metabolism, hypoxia, and TNFα signaling via NFκB gene sets were enriched. The mice demonstrated alteration of the gut microbiome, especially increase of family *Erysipelotrichaceae*, the bacteria that were reported to be enriched in patients with NAFLD compared to healthy subjects [59]. Metagenome prediction in the gut microbiota showed enriched citrate cycle and carbon fixation pathways [60].

### 4.4. Oral Administration Model

Mice orally administered *P. gingivalis* twice a week for 5 weeks experienced increases in insulin resistance and systemic inflammation. Oral administration of *P. gingivalis* caused an increase in hepatic fat and triglyceride levels as well as the mRNA expression of *TNF-α*, *IL-6*, *Fitm2*, and *Plin2*, with the latter two being strongly associated with lipid droplet formation in the liver. In addition, mRNA expression of *Acaca* and *G6pc*, which positively regulate fatty acid synthesis and gluconeogenesis, respectively, were also upregulated. Blood endotoxin levels tended to be higher, whereas gene expression of tight junction proteins in the ileum was significantly decreased in *P. gingivalis*-administered mice. Furthermore, pyrosequencing revealed that the population belonging to *Bacteroidales* was significantly increased in the gut of *P. gingivalis*-administered mice [61].

Mice fed on high-fat diet for 6 weeks and administered *A. actinomycetemcomitans* 6 times a week also presented impaired glucose tolerance and insulin resistance compared to saline-administered control mice. Oral administration of *A. actinomycetemcomitans* increased liver steatosis and enriched glucagon signaling pathway, adipocytokine signaling pathway, and insulin resistance in the liver. Based on 16S rRNA sequencing, *A. actinomycetemcomitans* administration changed the composition of the gut microbiota, especially decreasing the genus *Turicibacter*, which correlates with the production of butyric acid [62]. An increase in butyrate levels has been associated with improved insulin sensitivity [63]. Therefore, administration of *A. actinomycetemcomitans* may affect insulin resistance by decreasing butyrate levels. Metagenome prediction in gut microbiota showed upregulation of fatty acid biosynthesis and downregulation of fatty acid degradation in *A. actinomycetemcomitans*-administered mice [9].

### 4.5. Ligature-Induced Periodontitis Model

Silk ligature is used to induce periodontitis in mice and rats. Ligature ligation around teeth causes local accumulation of anaerobic bacteria and periodontal tissue represents rapid bone loss [64,65,66,67].

The livers of rats with ligature-induced periodontitis showed extensive microvesicular steatosis with reduced NG2-positive pericytes. Although serum levels of ALT and AST did not differ between control and ligated rats, periodontitis induced a significant decrease of GSH and increased MDA concentrations in the liver [68]. Rats were fed a high-fat diet for 12 weeks and the ligature was placed around the maxillary first molar tooth at 4 weeks. *P. gingivalis* slurry was applied around the ligature twice a week for 8 weeks. The rats showed NASH characterized by perivenular lipid deposition, including large fatty drops, ballooning degeneration, and focal necrosis with inflammation. Moreover, significant increases in alanine aminotransaminase, AST, C-reactive protein, and endotoxin levels were observed in the periodontitis rats with *P. gingivalis* [69].

**Table 5 ijms-22-03728-t005:** In vivo basic research regarding the relationship between NAFLD and periodontal disease.

Ref. No.	Study	Animals	High-Fat Diet	Model	Major Findings
[8]	Yoneda et al.	Mice	+	Intravenous injection of *P. gingivalis*	Increase in the body weight acceleration in the progression of NAFLD
[9]	Komazaki et al.	Mice	+	Oral administration of *A. actinomycetemcomitans*	Increased liver steatosis the enriched glucagon-signaling pathway, adipocytokine signaling pathway, insulin resistance in the liver decrease in the genus *Turicibacter* in the gut.
[54]	Nakahara et al.	Mice	+	Pulp chamber model	Fatty acid metabolism was disrupted, and expression levels of SCD1 and ELOVL6 were reduced.
[55]	Furusho et al.	Mice	+	Pulp chamber model	*P. gingivalis* was detected in Kupffer cells and hepatocytes
[56]	Fujita et al.	Rats	+	*P. gingivalis* LPS injection in gingiva	Large fat droplets Ballooning degeneration Infiltration of inflammatory cells
[57]	Varela-López et al.	Rabbits	+	*P. gingivalis* LPS injection in gingiva	High score of acinar inflammation Increase in the blood triglyceride and phospholipid levels
[58]	Nagasaki et al.	Mice	+	Pulp chamber model	Ipregulation of the immunoexpression of phosphorylated Smad2 and Galectin-3
[60]	Sasaki et al.	Mice	+	Intravenous injection of *P. gingivalis*	Impaired glucose tolerance, insulin resistance, and liver steatosis Alteration of the gut microbiome
[61]	Arimatsu et al.	Mice	-	Oral administration of *P. gingivalis*	Increase in insulin resistance and systemic inflammation Increase in the order *Bacteroidales* in the gut
[68]	Vasconcelos et al.	Rats	-	Ligature-induced periodontitis model	Decrease of GSH and increase of MDA concentrations
[69]	Kuraji et al.	Rats	+	Ligature-induced periodontitis model	Perivenular lipid deposition, including large fatty drops, ballooning degeneration, and focal necrosis with inflammation.

NAFLD: nonalcoholic fatty liver disease, LPS: lipopolysaccharides, GSH: glutathione, MDA: malondialdehyde.

## 5. In Vitro Basic Research Regarding the Relationship between NAFLD and Periodontal Disease

Only a few studies have shown the effect of periodontal disease on NAFLD in vitro. The contents of this section are summarized in Table 6. Human hepatocellular cells (HepG2) accumulated more intracellular lipids when stimulated with *P. gingivalis* LPS compared to cells treated with *Escherichia coli* LPS or the control not treated with LPS. Moreover, *P. gingivalis* LPS treatment significantly upregulated MyD88 and proinflammatory cytokines and increased the phosphorylation of p65 and JNK in HepG2 cells. Suppression of the phosphorylation of p65 and JNK by inhibitors and shRNA reduced lipid accumulation upon *P. gingivalis* LPS stimulation, suggesting that *P. gingivalis* LPS might contribute to intracellular lipid accumulation and inflammation in HepG2 cells via the activation of the NF-κB and JNK signaling pathways [70]. In addition, oleic acid-induced HepG2 cells as an in vitro model for NAFLD showed an increase in the presence of *P. gingivalis* in the cells at an early phase of infection. Lipid droplets affected the removal of *P. gingivalis* by altering the autophagy-lysosome system [71]. Palmitate-treated Hc3716-hTERT as an in vitro steatotic hepatocyte model showed upregulation of TLR2 expression. *P. gingivalis* LPS stimulation increased the mRNA levels of inflammasomes and proinflammatory cytokines in steatotic hepatocytes [55]. *P. gingivalis* infection markedly induced TGF-β1 and Galectin-3 production in LX-2 and Hc3716-hTERT cells [58].

## 6. Immunological Responses in Periodontitis and NAFLD

### 6.1. Role of T Cells in Periodontal Disease

In 1976, Page et al. showed that activation of the adaptive immune system is important for the progression of periodontitis by histological analysis of specimens from patients with periodontal disease [72]. Previous studies of experimental periodontal disease suggested that adaptive T cell immunity was required for alveolar bone destruction, as T cell deletion resulted in disease resistance [73]. Subsequent analysis using mice showed that CD4-positive T cells played an important role in the destruction of periodontal tissue [74]. Previous studies have focused specifically on the role of Th1/Th2 cells in periodontal disease. Experimental periodontal disease studies have shown that IFN-γ-deficient mice are more resistant to alveolar bone loss after oral infection by periodontal pathogens, such as *P. gingivalis* [75]. However, in IFN-γ-deficient mice, an increase in the number of bacteria was observed in the periodontium, the acute phase reaction after infection became stronger, and the mice died due to disseminated bacterial infection [76]. These results indicate that the Th1 cytokine IFN-γ acts suppressively against oral bacterial infections but may promote the regulation of bone resorption due to periodontal disease. On the other hand, IFN-γ strongly suppresses osteoclast differentiation, and there may be pathological conditions that cannot be explained by Th1/Th2 balance [77,78,79]. Th17 cells are a defined subpopulation of T helper cells named after IL-17A, a cytokine that is primarily secreted [80]. In addition to IL-17A, members of the IL-17 family include IL-17B, IL-17C, IL-17D, and IL-17E [81,82]. Of these, IL-17F shares the greatest homology with IL-17A, and the two cytokines may be regulated by similar mechanisms due to chromosomal proximity and coordinated expression patterns. IL-17A is the most widely studied member of the IL-17 family and mediates many of the known effector functions of Th17 cells [83,84,85]. IL-17A is involved in host protection from certain extracellular bacteria and fungi, such as *Klebsiella pneumoniae* and *Candida albicans* [86,87]. IL-17A induces many proinflammatory cytokines, such as IL-6, granulocyte colony stimulating factor, and TNF-α in both immune and non-immune cells. Among other activities, it leads to the activation of innate immune cells, proinflammatory signaling pathways, and neutrophil recruitment. Therefore, IL-17A is an important factor that plays a protective role against infection. The presence of IL-17A-producing cells in human and mouse gingival tissues has been demonstrated using immunohistochemistry and flow cytometry [88,89]. Analysis using mice revealed that Th17 cell accumulation in periodontal disease tissues is dependent on oral bacteria. In human studies, IL-17A-producing cells were more abundant in the gingival tissue of patients with periodontal disease than in healthy gingival tissue, resulting in infiltration of IL-17A-producing cells and inflammation of periodontal disease tissue [90]. This suggests that the number of Th17 cells may correlate with the degree of gingival inflammation. Interestingly, Tsukasaki et al. reported that mice lacking both IL-17A and IL-17F were found to not only suppress periodontal disease-related bone destruction but also increase the amount of bacteria in periodontal tissue. Furthermore, the composition of indigenous bacteria in the oral cavity was also changed [89].

This suggests that Th17 cells contribute to both the destruction of alveolar bone and elimination of oral bacteria. In fact, Th17-related cytokines contribute to the maintenance of the intestinal flora and the secretion of antibacterial peptides, and may have a similar role in indigenous bacteria in the oral cavity [90,91].

### 6.2. Innate Immune Response in NAFLD

In NAFLD, Kupffer cells and monocyte-derived macrophages are the major players in innate immunity [92]. In the steady state, Kupffer cells can inhibit dendritic cell-induced antigen-specific T cell activation and promote regulatory T cell (Treg) inhibitory activity [93]. The main action of Tregs is to prevent self-reaction to self-antigens. It can also avoid excessive effector T cell activation and subsequent tissue damage during an induced immune response [94]. The role of Tregs in NAFLD is described below. Kupffer cells secrete CCL2 and other substances to mobilize monocytes to the liver, and inflammatory cytokines, such as TNF-α, promote liver fibrosis [95].

### 6.3. Role of T Cells in NAFLD

Regarding acquired immunity, T cells are particularly involved in NAFLD. Among T cells, there are cell types that exert pathogenicity and those that act defensively. Tregs are anti-inflammatory cells that use the transcription factor Foxp3 as a master regulator [94]. As mentioned above, since Tregs play a role in suppressing autoimmunity, Treg deficiency shows a lethal phenotype in both mice and humans [96,97,98]. Tregs present in visceral adipose tissue suppress the inflammation of visceral adipose tissue and maintain constant insulin sensitivity and glucose tolerance. In fact, animal models of obesity have reduced Treg numbers in the spleen and are negatively associated with insulin resistance [99]. The decrease in Tregs is due to their apoptosis by reactive oxygen species, and the adoption of Tregs reduces inflammatory cytokines and suppresses steatohepatitis [100]. As a cell type showing pathogenicity, Th1 cells are associated with the inflammation of adipose tissue [101,102]. Th1 cell-deficient mice and IFN-γ-deficient mice show reduced adipose tissue inflammation and improved glucose tolerance [102,103]. In contrast, Th1 cells reduce IFN-γ production when co-cultured with stem cells derived from adipose tissue in obese patients [104]. These discrepancies could not be explained by the Th1/Th2 paradigm. In recent years, the association between NAFLD and Th17 has attracted attention [105,106,107]. Th17 cells are abundant in the liver and peripheral blood of animal models of NAFLD. Studies involving humans have shown increased Th17 levels in the adipose tissue and peripheral blood in patients with obesity and type 2 diabetes, and patients with NASH have shown increased Th17 levels, similar to mice [108]. However, the exact role of Th17 cells in the development of steatosis is currently unclear. In an analysis using mice, IL-17A-, IL-17F-, and IL-17RA-deficient mice showed liver steatosis in a fatty liver model due to a methionine choline-deficient diet, but the degree of liver dysfunction was significantly lower than that of wild-type mice [109]. However, in terms of liver fibrosis, several studies have reported that the Th17 axis is pathogenic; thus, it is speculated that liver fat accumulation and liver fibrosis are caused by different factors [110,111].

### 6.4. Interrelationship between NAFLD and Periodontal Disease

Many studies have gradually revealed the relationship between periodontal disease and NAFLD. Examining the relationship from an immunological point of view, the Th17 axis in vivo may be activated by infection with *P. gingivalis*. On the other hand, obesity is induced by lifestyle factors, such as eating habits and decreased exercise, and liver steatosis is promoted. The Th17 axis is increasingly activated according to the progression of liver steatosis and may affect the progression of hepatitis and fibrosis. As a treatment for suppressing the Th17 axis, IL-17A and IL-23 inhibitors have been clinically applied as therapeutic agents for inflammatory bowel disease [112] and psoriasis [113], and these drugs may be effective against NAFLD and periodontal disease. Oral pathobiont-reactive Th17 cells arose from periodontal inflammation could migrate to the inflamed gut, and could be activated by translocated oral pathobionts and cause development of colitis [114]. Th17 cells induced by periodontitis may migrate to the liver and aggravate NAFLD.

## 7. Conclusions

This review summarizes the relationship between NAFLD and periodontal disease (Figure 1) and shows the possibility that periodontal disease aggravates NAFLD. Epidemiological studies have revealed positive associations between periodontal disease and the onset/progression of NAFLD. Furthermore, an in vivo animal model showed a piece of the mechanism by which periodontal disease affects NAFLD. Detection of periodontopathic bacteria in the liver suggested that bacteremia caused by periodontal disease has a direct influence on NAFLD. Remarkably, alterations in the gut microbiome were also observed after the administration of periodontopathic bacteria. Although the number of in vitro studies is limited, some have shown that *P. gingivalis* LPS induces inflammation and intracellular lipids in hepatocytes. In addition, Th17 might be a key molecule for explaining the relationship between periodontal disease and NAFLD.

Successful periodontal treatment is effective for glycemic control in patients with type 2 diabetes and periodontal disease [14,115,116]. However, the effects of periodontal treatment on NAFLD are currently unclear. However, the progression and aggravation of NAFLD and type 2 diabetes are closely related. Further studies are required to clarify the relationship between periodontal disease and NAFLD and to establish the general concept that oral health is essential for systemic health, especially in patients with NAFLD.

## Figures and Tables

**Figure 1 ijms-22-03728-f001:**
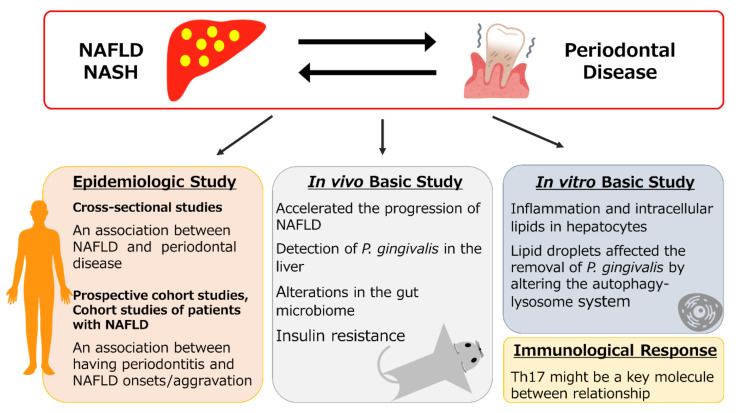
The summary of this review. Periodontal disease has an association with NAFLD.

**Table 1 ijms-22-03728-t001:** Cross-sectional studies regarding the relationship between nonalcoholic fatty liver disease (NAFLD) and periodontal disease.

Ref. No.	Study Country Year Sample Size	Parameters Evaluated for the Diagnosis of NAFLD	Parameters Evaluated for the Diagnosis of Periodontitis	Major Findings
[34]	Furuta et al. Japan 2010 *n* = 2225	Serum ALT level	PD, BOP	An association between periodontitis and serum ALT in male without smoking (OR = 2.3, 95% CI = 1.0–5.2)
[35]	Ahmad et al. Japan 2017 *n* = 5683	Serum ALT level	PD, CAL	An association among deep periodontal pockets and combination of increased serum ALT and symptoms of metabolic syndrome in male with low alcohol consumption
[36]	Morita et al. Japan 2014 *n* = 1510	Serum GGT, ALT, AST level	CPI	An association between elevation of alglutamyltranspeptidase and having deep periodontal pockets (OR = 1.48, 95% CI = 1.16–1.90)
[37]	Iwasaki et al. Japan 2018 *n* = 1226	Ultrasonography	PD, BOP	An association between periodontitis and NAFLD (OR = 1.881, 95% CI = 1.184–2.987)
[38]	Kim et al. Korea 2020 *n* = 4272	FLI	CPI	An association between periodontal disease and FLI (OR = 1.63; 95% CI = 1.23–2.16)
[39]	Shin et al. Korea 2019 *n* = 4061	FLI, HSI	CPI	An association between periodontal disease and FLI, HSI in women (OR = 1.77, 95% CI = 1.05–2.98)
[40]	Qiao et al. China 2018 *n* = 24,470	Ultrasonography	The number of missing teeth	An association between the missing teeth and NAFLD in men (among those who with more than 6 missing teeth, OR = 1.40, 95% CI = 1.09–1.81)
[41]	Weintraub et al. USA 2019 *n* = 5421	Ultrasonography, Fibrosis Score, FLI, US-FLI	PD, BOP, CAL the number of missing teeth	An association between periodontitis, tooth loss and all of the parameters for NAFLD
[42]	Alazawi et al. USA 2017 *n* = 8172	Ultrasonography	PD, CAL Serum antibody titers against 19 oral bacteria	Significant associations among steatosis and bleeding on probing, PD ≥ 4 mm (%), mean PD, CAL ≥ 3 mm, and mean CAL (%) After adjusting for sociodemographic factors, only BOP and mean PD showed a significant association with steatosis (BOP: OR = 1.07, 95% CI = 1.00–1.14, mean PD: OR = 1.08, 95% CI = 1.00–1.17)
[43]	Akinkugbe et al. USA 2017 *n* = 11,914	Ultrasonography, serum ALT level	PD, CAL	Periodontitis was associated with serum ALT and AST levels (≥ 30% of sites with PD ≥ 4 mm: OR = 1.39, 95% CI = 1.02–1.90), however, the significance was not observed after adjustment of age and sex
[46]	Akinkugbe et al. Germany 2017 *n* = 2481	Ultrasonography	PD, CAL	A significant correlation between periodontitis and NAFLD among subjects with less than 1 mg/L serum CRP levels and/or with lower than the median weighted genetic CRP score Serum CRP modified the interaction between periodontitis and NAFLD

NAFLD: nonalcoholic fatty liver disease, ALT: alanine aminotransferase, FLI: Fatty Liver Index, HSI: Hepatic Steatosis Index, CPI: Community Periodontal Index, PD: pocket depth, CAL: clinical attachment level, BOP: bleeding on probing, CRP: C-reactive protein.

**Table 2 ijms-22-03728-t002:** Prospective cohort studies regarding the relationship between NAFLD and periodontal disease.

Ref. No.	Study Country Year Sample Size	Evaluation Criteria for Liver	Parameters Evaluated for the Diagnosis of Periodontitis	Observation Period	Major Findings
[47]	Kuroe et al. Japan 2020 *n* = 341	NAFL (ultrasonography, NAFLD fibrosis score)	PD, CAL	5 years	CAL and liver fibrosis were significantly associated in obese NAFL patients (OR = 2.87, 95% CI = 1.23–6.69)
[48]	Akinkugbe et al. Germany 2017 *n* = 2623	NAFLD (ultrasonography)	PD, CAL	median 7.7 years	NAFLD incidence was elevated in participants with >30% of ≥3 mm CAL (multivariable-adjusted incidence rate ratio: 1.60, 95% CI = 1.05–2.43)
[49]	Helenius-Hietala et al. Finland 2017 *n* = 1801	The incidence of severe liver disease (a first hospitalization owing to liver disease or liver-related death, a diagnosis of (primary) liver cancer)	PD	13 years	The incidence of severe liver disease was increased for the patients with advanced periodontitis and NAFLD (hazard ratio = 6.94, 95% CI = 1.43–33.6)

NAFLD: nonalcoholic fatty liver disease, PD: Procket depth, CAL: clinical attachment level.

**Table 3 ijms-22-03728-t003:** Meta-analysis regarding the relationship between NAFLD and periodontal disease.

Ref. No.	Study	The Number of Primary Studies	Study Designs of Primary Studies	Statistical Analysis	Results
[50]	Chen et al. 2020 *n* = 118,408	12	Cross-sectional (4), case-control (1), cohort (7)	Generalized least-squares regressions	An association between periodontitis and NAFLD (OR = 1.19, 95% CI =1.06–1.33), and an association periodontitis and cirrhosis (OR = 2.28, 95% CI = 1.50–3.48) was reported.
[51]	Wijarnpreecha et al. 2020 *n* = 27,703	5	Cross-sectional (4), cohort (1)	PRISMA, The random-effect model	NAFLD was associated with periodontitis (OR = 1.48, 95% CI = 1.13–1.89), however, significant correlation was lost after the adjusted results of the primary studies were applied.

NAFLD: nonalcoholic fatty liver disease, PRISMA: Preferred Reporting Items for Systematic Reviews and Meta-analysis.

**Table 4 ijms-22-03728-t004:** Studies regarding periodontal disease in patients with NAFLD.

Ref. No.	Study Country Year Sample Size	Parameters Evaluated for the Diagnosis of NAFLD	Parameters Evaluated for the Diagnosis of Periodontitis	Major Findings
[8]	Yoneda et al. Japan 2012 *n* = 150	Liver biopsy	Detection of *P. gingivalis* in saliva by PCR	The detection of *P. gingivalis* was higher in NAFLD patients compared to non-NAFLD control subjects (46.7% vs. 21.7%, OR = 3.16, 95% CI = 1.58–6.33). Serum AST and ALT levels of 10 patients with NAFLD were significantly decreased after receiving periodontal treatment for 3 months.
[9]	Komazaki et al. Japan 2017 *n* = 52	Total fat area, visceral fat area and the liver/spleen ratio evaluated by abdominal computed tomography, fasting blood insulin level, HOMA-IR, Serum AST, ALT, and γ-GTP	Serum antibody titers against *A. actinomycetemcomitans*, *F. nucleatum*, *P. gingivalis*	Anti-*A. actinomycetemcomitans* antibody titers and anti-*F. Nucleatum* antibody titers were slightly associated with fat area. Anti-*A. actinomycetemcomitans* antibody titers showed a positive correlation with fasting plasma insulin, the homeostasis model of assessment of insulin resistance, and AST, and a negative correlation with the liver/spleen ratio.
[42]	Alazawi et al. UK 2017 *n* = 69	Ultrasonography	PD, CAL	An association among periodontitis and the severity of NAFLD (NASH vs. NAFL) and the presence of diabetes was reported.
[54]	Nakahara et al. Japan 2018 *n* = 200	Liver biopsy	Serum antibody titers against *P. gingivalis*	A significant monotonic trend between the fibrosis stage and antibody titers against *P. gingivalis fim A* type 1 and 4 was observed. An association between antibody titers against *P. gingivalis* fim A type 4 and advanced fibrosis was reported (OR = 2.081, 95% CI = 1.098–3.943).
[55]	Furusho et al. Japan 2013 *n* = 40	Liver biopsy	Detection of *P. gingivalis* by immunohistochemistry in hepatocytes	*P. gingivalis*-positive patients showed progression of hepatic fibrosis compared to patients without *P. gingivalis*.

NAFLD: nonalcoholic fatty liver disease, PCR: Polymerase chain reaction, HOMA-IR: the homeostatis model of assessment of insulin resistance, AST: Aspartate transaminase, ALT: alanine aminotransferase, GTP: glutamyl transpeptidase.

**Table 6 ijms-22-03728-t006:** In vitro basic research regarding the relationship between NAFLD and periodontal disease.

Ref. No.	Study	Cells	Major Findings
[55]	Furusho et al.	Palmitic acid-induced Hc3716-hTERT cells	Upregulation of TLR2 expression Increase in the mRNA levels of inflammasomes and proinflammatory cytokines
[58]	Nagasaki et al.	Palmitic acid-induced LX-2 and Hc3716-hTERT cells	Induction of TGF-β1 and Galectin-3 production
[70]	Ding et al.	oleic acid-induced HepG2 cells	Accumulation of intracellular lipids Enhancement in the phosphorylation of p65 and JNK
[71]	Zaitsu et al.	oleic acid-induced HepG2 cells	Lipid droplets affected the removal of *P. gingivalis* by altering the autophagy-lysosome system

NAFLD: nonalcoholic fatty liver disease, hTERT: Human telomerase reverse transcriptase, HepG2: Human Hepatocellular Carcinoma.

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
