# Peer review of "Relationship between NAFLD and Periodontal Disease from the View of Clinical and Basic Research, and Immunological Response"

_ijms, 2021, doi:10.3390/ijms22073728_

Round 1

Reviewer 1 Report

This manuscript is well written and is easy to read. I have only minor suggestions.

  1. The authors mainly described the effects of periodontal disease on nonalcoholic fatty liver disease (NAFLD). On the other hand, what kind of effects does NAFLD have on periodontal health?

  1. What is the clinical significance of NAFLD accompanied with periodontal disease? Please clarify this point.

  1. Does the relationship between NAFLD and periodontal disease differ according to the severity of periodontal disease?

  1. (Page 6, Ref no. 40 in Table 1) Change the words from “PPD” to “PD”.

Author Response

We thank the reviewer for their constructive and encouraging comments. We believe that incorporation of all the reviewers’ suggestions has significantly improved the revised manuscript. Significant changes are in red in the revised manuscript file. Below, we respond to each question and concern and note changes that have been made. We hope that our responses address your concerns satisfactorily.

Responses to Reviewer 1

The authors mainly described the effects of periodontal disease on nonalcoholic fatty liver disease (NAFLD). On the other hand, what kind of effects does NAFLD have on periodontal health?

We thank the reviewer for the comment. Most of the researches focused on the effect of periodontitis on NAFLD, not mutual interaction between periodontitis and NAFLD, nor the effect of NAFLD on periodontitis. Therefore, the effect of NAFLD on periodontitis remains unclear. However, there is a prospective study which investigated the effect of NAFLD on the aggregation of periodontitis (ref 48). According to the research, NAFLD may contribute to the aggregation of periodontitis. We added following sentence about the detail of the research.

“Surprisingly, the incidence rate ratio of the change in CAL in 5 years significantly increased upon combination with NAFLD in participants with > 30% of ≥ 3 mm CAL, whereas the significance was not observed after adjustment for confounding factors. NAFLD may contribute to the aggravation of periodontitis.” (page 7, lines 209-213)

Also, we have corrected the word “periodontal disease” to “periodontitis” in the description of the same research to be more precise. (page7, line 207)

What is the clinical significance of NAFLD accompanied with periodontal disease? Please clarify this point.

Direct impact of periodontal disease on the clinical pathogenesis of NAFLD has not been fully understood. Available evidences showed the association between the infection of periodontitis related bacteria and the hepatic phenotypes of human NAFLD. Correlation and multivariate analysis identified that titer of blood antibody of periodontitis-related bacteria related to liver steatosis and fibrosis. In order to clarify the significant and direct impact of periodontitis disease on human NAFLD, further study will be needed. Accordingly, we added the description below in the end of “NAFLD” section (page 3, lines 96-102).

“The direct effects of periodontal disease on the clinical pathogenesis of NAFLD and NASH are not fully understood. However, multiple cross-sectional studies have shown the association between infection by periodontitis-associated bacteria and the hepatic phenotypes of human NAFLD. That said, the significance of periodontitis in the clinical outcomes of NAFLD and NASH, including mortality, liver cancer, liver failure, and cardiovascular disease events, remains unclear. A longitudinal randomized controlled study should be conducted for evaluating the effect of periodontitis treatment on NAFLD and NASH in humans.”    

Does the relationship between NAFLD and periodontal disease differ according to the severity of periodontal disease?

We thank reviewers for the suggestion to focus on the severity of periodontitis. There are four cross-sectional studies (Ref 37, 38, 39, 46), and two prospective studies (Ref 48, 49) analyzing based on the severity of periodontitis. We described the major findings of those researches in more detail by adding these sentences:

Ref 37: “In all participants and in female participants, the prevalence of NAFLD increased with the increase in periodontal disease severity.” (page 4, lines 149-151)

Ref 38: “A stronger association was observed between FLI and periodontitis prevalence in the diabetes subgroup. In addition, patients with diabetes showed more severe periodontitis.” (page 4, lines 153-155)

Ref 39: “The severity of periodontal disease was positively associated with FLI in female participants.” (page 4, lines 155-156)

Ref 46: “Subjects with extensive periodontitis (≥ 30% sites with PD ≥ 4 mm) had higher odds ratios for NAFLD than patients with moderate periodontitis (< 30% sites with PD ≥ 4 mm) among participants with a serum CRP level < 1 mg/L.” (page 5, lines 189-192)

Ref 48: “However, NAFLD and periodontitis were not associated in participants with < 30% affected sites.” (page 7, lines 209-210)

Ref 49: “The hazard ratio for the incidence of severe liver disease increased with the severity of periodontitis.” (page 7, lines 217-218)

We summarized the influence of the severity of periodontitis on NAFLD by adding this sentence in the concluding part of epidemiologic studies.

“In addition, compared to moderate periodontal disease, severe periodontal disease is more strongly associated with NAFLD.” (page 5, lines 194-195)

(Page 6, Ref no. 40 in Table 1) Change the words from “PPD” to “PD”.

Thank you for pointing it out. We have changed the words in the Table 1 and manuscript (page 5, line 172) according to the reviewer’s suggestion.

Reviewer 2 Report

A well designed review article on a current topic, the relationship between NALFD and periodontitis.

Some minor corrections needed:

"2.2 From NAFLD to metabolic associated fatty liver disease (MAFLD)"

 Too much emphasis on nomenclature, not enough on the physiopathological aspects. Please add more information from the literature.

Table 1. 41 Akinkugbe et al. Periiodontitis was associated with

Author Response

We thank the reviewer for their constructive and encouraging comments. We believe that incorporation of all the reviewers’ suggestions has significantly improved the revised manuscript. Significant changes are in red in the revised manuscript file. Below, we respond to each question and concern and note changes that have been made. We hope that our responses address your concerns satisfactorily.

Responses to Reviewer 2

"2.2 From NAFLD to metabolic associated fatty liver disease (MAFLD)"

 Too much emphasis on nomenclature, not enough on the physiopathological aspects. Please add more information from the literature.

MAFLD is recently advocated by Eslam et al. in 2020 [30]. Therefore, evidence related with MAFLD has not been well accumulated and epidemiology, pathogenesis and association with comorbid disease including periodontal disease are unclear. On the other hand, considering the diagnostic criteria of MAFLD which focuses on the metabolic dysfunction rather than NAFLD, association with periodontal disease of MAFLD might be more significant, and might be more complicated than NAFLD. Regarding with the literature by Eslam et al. and previous reports, we inserted following discussion in the section 2.2 (page 3, lines 114-126).

“MAFLD can be diagnosed if either obesity or type 2 diabetes is diagnosed concomitantly with fatty liver disease. MAFLD can also be diagnosed in lean or non-obese patients with fatty liver if they have at least two metabolic abnormalities, including dyslipidemia, hypertension, insulin resistance, and prediabetes. Moreover, other chronic liver diseases, such as viral hepatitis, autoimmune diseases, or alcoholic liver injury, are not an exclusion criteria for MAFLD diagnosis [30]. To the best of our knowledge, the association between MAFLD and periodontal disease has not been reported previously. Considering the broader spectrum of MAFLD, which focuses on metabolic abnormalities and can occur in patients with any other chronic liver disease, the association between periodontal disease and MAFLD may be significant and more complicated than that with NAFLD. Type 2 diabetes and metabolic syndrome are the sole risk factors for periodontal disease [31], and the pathogenesis of viral hepatitis and alcoholic liver injury have also been linked to periodontal disease [32]. Further studies are necessary to identify the clinical significance of periodontal disease in patients with MAFLD.“ 

Table 1. 41 Akinkugbe et al. Periiodontitis was associated with

Thank you for pointing it out. We have corrected the word to Periodontitis in the Table 1.